# Simulating Urban Growth Using the Cellular Automata Markov Chain Model in the Context of Spatiotemporal Influences for Salem and Its Peripherals, India

Linda Theres [1,†] , Selvakumar Radhakrishnan [1,*] and Abdul Rahman [2,†]

1 School of Civil Engineering, SASTRA Deemed to be University, Thanjavur 613401, Tamil Nadu, India; lindatheres@civil.sastra.ac.in
2 Department of Geography and Geoinformatics, Bangalore University, Bengaluru 560056, Karnataka, India
* Correspondence: selvakumar@civil.sastra.edu
† These authors contributed equally to this work.

**Abstract:** Urbanization is one of the biggest challenges for developing countries, and predicting urban growth can help planners and policymakers understand how spatial growth patterns interact. A study was conducted to investigate the spatiotemporal dynamics of land use/land cover changes in Salem and its surrounding communities from 2001 to 2020 and to simulate urban expansion in 2030 using cellular automata (CA)–Markov and geospatial techniques. The findings showed a decrease in aerial vegetation cover and an increase in barren and built-up land, with a rapid transition from vegetation cover to bare land. The transformed barren land is expected to be converted into built-up land in the near future. Urban growth in the area is estimated to be 179.6 sq km in 2030, up from 59.6 sq km in 2001, 76 sq km in 2011, and 133.3 sq km in 2020. Urban sprawl is steadily increasing in Salem and the surrounding towns of Omalur, Rasipuram, Sankari, and Vazhapadi, with sprawl in the neighboring towns surpassing that in directions aligned toward Salem. The city is being developed as a smart city, which will result in significant expansion and intensification of the built-up area in the coming years. The study's outcomes can serve as spatial guidelines for growth regulation and monitoring.

**Keywords:** urbanization; geospatial; support vector machine; transition matrix; validation; sustainability

## 1. Introduction

Rapid and unplanned urban sprawl causes issues such as landscape fragmentation, decreased arable land, increased urban poverty, and environmental degradation. The United Nations estimates that urban areas will encompass 60 percent of the world's rural populations by 2050. In developing countries such as India, the population in medium-sized urban areas with less than 1 million people has increased significantly [1]. One of the objectives of the United Nation's Sustainable Development Agenda is to ensure safe, resilient, and sustainable cities by 2030. Policies to promote the sustainable development of cities, especially in developing nations, require precise and timely monitoring and understanding of urban growth. Urban models serve as a quantitative tool for urban and environmental planning, capability management, and land suitability for development [2]. Although the models' extent of applicability is broad, their ability to analyze spatial relationships is limited. Geospatial techniques provide a better environment for spatially addressing the issue at hand. Resource satellites and image processing tools have made urban sprawl modeling more realistically achievable. GIS, on the other hand, uses satellite technology to monitor and model geographically referenced morphological changes in metropolitan areas.

Rapid urbanization and resulting land use and cover changes have become a primary global concern in natural resource management and sustainable development worldwide.

Since the 1960s, researchers have discussed models such as cellular automata [3], the artificial neural network [4], the Markov chain [5], geographically weighted regression [6], the non-ordinal and Sleuth model [7], the analytic hierarchy process [8], machine learning models [9] and an urban sprawl matrix [10] to analyze the growth and patterns. Markov chain analysis is the best method for modeling LU/LC changes, especially when transitions and processes in the landscape are difficult to characterize [11]. The Markov chain is a time series model based on machine learning that forecasts change in LU/LC based on prior rates of change [12]. The model, however, predicts future LU/LC changes in the temporal dimension but not in the geographical dimension [13]. In contrast to the Markov chain model, the cellular automata (CA) model has a spatial component in which specific rules from surrounding cells anticipate future change [14]. Simulating urban dynamism using remote sensing, GIS, and basic transition rules is simple. Despite its advantages, CA's technical infrastructure is insufficient to deal with real-world urban dynamics. Furthermore, it will not consider the impact of external forces on urban expansion. On the other hand, it has been proposed that incorporating Markov chain analysis into CA models will help to overcome these limitations [15,16]. Cellular automata (CA)–Markov models have been widely used in urban studies, particularly in simulating and understanding urban dynamics, transportation network developments, and planning infrastructure [17,18]. The application of the CA–Markov models has yielded several significant findings. For instance, studies have shown that the availability of infrastructure and the expansion of transportation networks can significantly impact urban sprawl and the conversion of agricultural land to urban use [19]. Similarly, the effects of population growth and income on urban expansion have been studied, with results indicating that income levels are critical in determining urban growth patterns. Despite their benefits, the application of CA–Markov models in urban studies has several gaps. Firstly, more research on the calibration and validation of these models is needed, particularly in areas with limited data [20]. Secondly, the impact of local policy interventions on urban growth patterns and land-use changes needs to be investigated further [21]. The study here aimed to address the first gap in the implementation of the model.

Driver variables are commonly used in CA–Markov models to simulate urban growth and land-use change. However, a simple CA–Markov model can still shed light on urban dynamics. Without driver variables, they are more straightforward, making models simpler to comprehend and communicate to the public and policymakers. Furthermore, they will be helpful when it is impractical or impossible to implement complicated models with many parameters. Another benefit of employing CA–Markov models without driver variables is the ability to create a baseline scenario for urban growth and change that can be compared to scenarios that include driver variables. This can provide light on the linkages between various urban growth drivers and help determine the relative relevance of various factors.

Dynamic urban growth simulation studies have typically focused on including driver variables to increase accuracy [22]. Even with the inclusion of driving variables in the model, the majority of studies acquire an accuracy of 0.68 kappa [23], 0.78 [24], 0.75 Kappa standard [25], and 0.79 kappa coefficient (kappa = correlation between two variables) [26]. Furthermore, obtaining the dataset for depicting drivers and integrating it into the model is a real challenge. It is possible to achieve significant results by incorporating the neighboring state and by limiting LU/LC changes [26]. This hypothesis is the foundation for the present study to determine whether desirable accuracy can be obtained using spatiotemporal changes and constraints as input into the model.

The role of India's smaller and medium-sized cities is projected to grow significantly by the year 2030 [27]. It is time that tier II cities became the primary focus of attention. As a result, Salem was chosen as the area of focus. Salem, a two-tier city in Tamil Nadu, has seen a rapid increase in human population since 1901, when it was 70,621, and it has increased nearly twelvefold to 829,267 in 2011. Rural migration and small-scale industry activity in Salem town contribute significantly to its urban growth. Additionally, Salem is one of Tamil Nadu's upcoming smart cities. Consequently, it will be established as a commercial and

industrial hub with a diverse economic base, providing equitable opportunities to all and resulting in increased agglomeration and sprawl. As a result, there is a need to concentrate on studying and projecting urban growth in this region.

Several researchers limit their sprawl studies to the city boundary [28–34]. However, the city boundary is not permanent and is susceptible to alteration according to the sprawl rate. The authors in [35] advocate defining a circular buffer around the city center, although there is no evidence to justify the radius. Sprawl occurs when individuals move to a city from outside the area. In the case of non-metropolitan cities, most of these migrations are from nearby towns; thus, the study region needs to be established by embracing the major towns surrounding the city under study. Therefore, this paper discusses a study that uses the CA–Markov model to predict sprawl in Salem city and its environs without relying on driving variables. It also reveals the proportional impact of geographical and temporal changes in LU/LC on urban sprawl in our region of interest.

## 2. Study Area

The city chosen as a region of interest is Salem, a nonmetropolitan city listed as a to-be smart city. It is located at 11°39′13″ N latitude and 78°09′12″ longitude, surrounded by Nagarmalai to the north, Jeragamalai to the south, Kanjanamalai to the west, and Godumalai to the east. It is an upland plateau and undulating terrain with a gentle slope to the east. The Thirumanimuthar river flows through the city and divides it into two. The southern and western sides of town are primarily plain agricultural lands with a few irrigation tanks. The Palar River constitutes the northern boundary of the town. The northeast monsoon significantly contributes to rainfall; the average annual precipitation ranges from 800 to 1600 mm. With such favorable living conditions, the city continues to expand; the encroachments are also gradually consuming more and more farmland. Salem's smart city plan must include proper monitoring and controlled expansion of urban sprawl.

Most urban sprawl studies focus on the city boundary extent, but it is not sensible because it is not constant, unlike other administrative boundaries. Aside from the dynamics of city boundaries, there will always be influences from nearby urban agglomerations. As a result, there is a need to broaden the scope of the research. The Salem City Municipal Corporation manages the city's civic administration and urban infrastructure. Salem is one of the 100 cities selected under the national Smart Cities Mission, with the objective of providing core infrastructure and improving the quality of life for citizens. The city has experienced a 2% population growth during 2001–2011, with most of it concentrated in the outskirts, and it supports a significant floating population from the surrounding areas [28]. Three national highways pass through the city, connecting it with major towns in the vicinity. Salem serves as the economic hub for these emerging towns located at varying distances along major highways. Hence, these towns, namely Omalur to the north, Rasipuram to the south, Sankari to the southwest, and Vazhapadi to the east, were included in the research. As shown in Figure 1, the extent of the boundary covering an area of 3891.68 sq. km has been considered for the study. The Yercaud forest in the northeast and a few other forests, such as the Jarugumalar reserved forest and the Nayinarmalai forest in the southwest, serve as a breakpoint for urban sprawl from Salem.

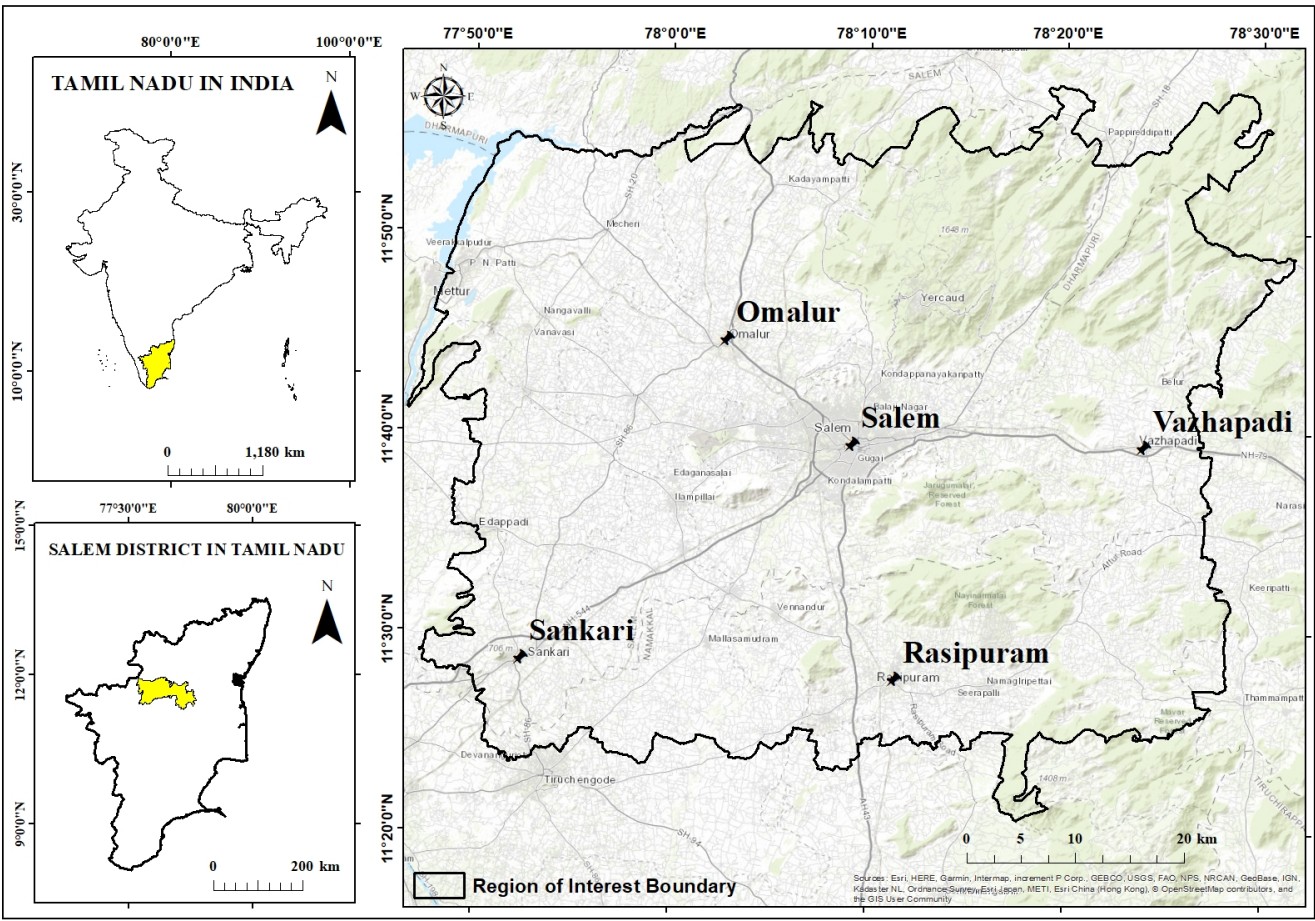

**Figure 1.** Map showing study area limit over a base map.

## 3. Data and Software

The satellite images used in this study for mapping land use features are Landsat 8 OLI (Operational Land Imager) data for 2020 and Landsat 7 ETM+ (Enhanced Thematic Mapper Plus) data for time slices 2011 and 2001 (Table 1). The Level 1 images were downloaded from Earth Explorer (https://earthexplorer.usgs.gov/, 18 March 2023), georeferenced in WGS84, and projected in UTM (UTM, zone 44 North). QGIS was used to perform atmospheric correction, and ERDAS Imagine 2014 was used for other preprocessing procedures. To generate LU/LC, the support vector machine algorithm was used in the Google Earth Engine code editor with JavaScript. Markov, Cell Atom, CA–Markov, and Validate module of Terrset software were used to run prediction modeling. The area of each class was computed, and the accuracy was assessed in the ArcGIS environment.

**Table 1.** Details of the satellite datasets used in this study.

| Year of Study | Satellite | Acquisition Date | Path/Row | Spatial Resolution |
|---|---|---|---|---|
| 2001 | Landsat 7 ETM+ | 09 December 2001 | 143/052 | 30 m |
| 2011 | Landsat 7 ETM+ | 19 January 2011 | 143/052 | 30 m |
| 2020 | Landsat 8 OLI | 20 January 2020 | 143/052 | 30 m |

## 4. Methodology

Figure 2 depicts the flow of processes involved in the study. The support vector machine algorithm was used to classify preprocessed Landsat data for three years: 2001, 2011, and 2020. The accuracy of datasets was then assessed using various parameters, followed by CA–Markov modeling for prediction.

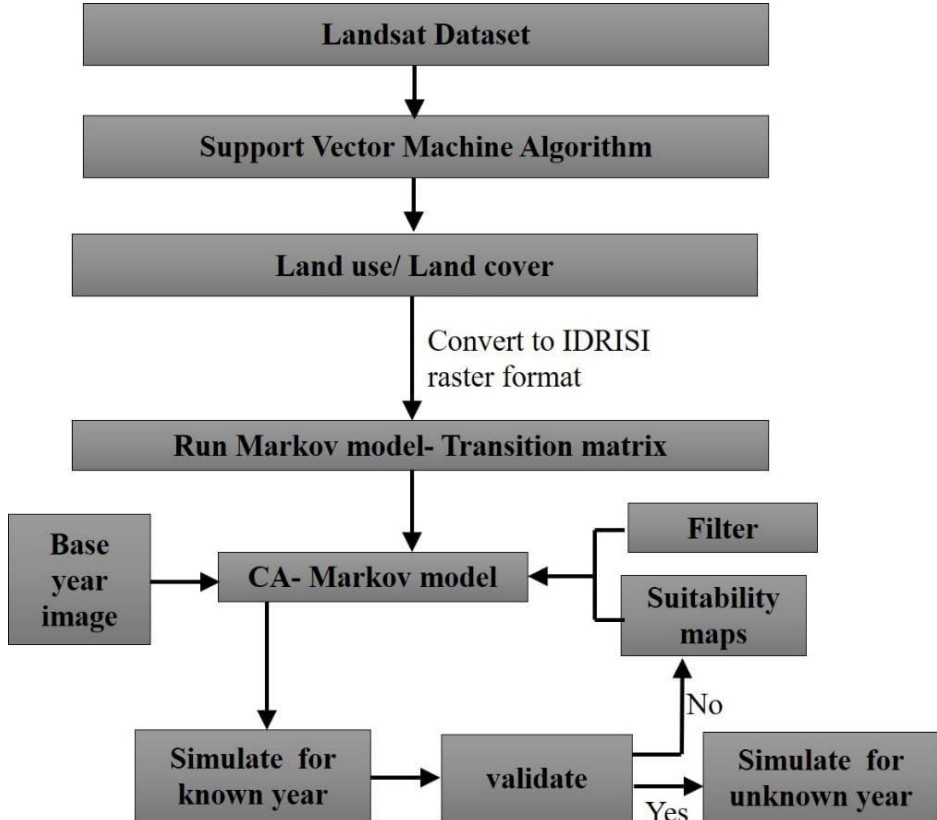

**Figure 2.** Methodology flowchart.

*4.1. Data Preprocessing*

Using the Semi-Automatic Classification Plugin (SCP) tool in QGIS, the atmospheric correction was applied to both the Landsat 7 and 8 datasets. At the same time, scan line correction was performed before atmospheric correction for the Landsat 7 (2011) dataset. Then, using layer stacking of bands, a false-color composite was created, which was then subset to the study area boundary. Pan-sharpening was used after removing no data value.

To enhance spatial resolution, pan-sharpening can be performed using resolution merge. Landsat tiles are provided with a multi-spectral sensor (MSS) and panchromatic (PAN) datasets, making it easy to pan-sharpen. The pan-sharpening method is determined by the input image and the purpose for which it is used [36]. The emphasis in our case is on delineating built-up structures. After several trials, a principal component (PC)-based pan-sharpening method was chosen. Regarding resampling methods, cubic convolution produced images with distinct boundaries compared to others that produced blurred boundaries. Pan-sharpening was performed in ERDAS IMAGINE using the resolution merge tool.

*4.2. LU/LC Classification*

Machine learning has significantly developed for LU/LC classification in recent years. Most of the research focuses on support vector machine (SVM) classification [37–41]. Although there is no optimal classifier, a comparative study based on performance indicators will aid in preparing a refined dataset [42–44]. Researchers compared different machine learning classifiers for the pan-sharpened Landsat dataset and revealed that SVM provided higher precision [45]. Accordingly, the support vector machine classifier was used to generate the land use/land cover (LU/LC) datasets. The algorithm is based on the concept of defining hyperplanes that serve as class boundaries. There can be n number of hyperplanes, of which the optimum one must be decided. The algorithm uses the nearest training point to the plane, also known as support vectors. The user-defined kernel is another parameter

used to address the generalization problem. The algorithm is run in the Google Earth engine using Java script for all three years (2001, 2011, and 2020), and the classified image is exported for further processing.

To capture land changes among major land types, six classes were considered: agricultural, forest, built-up, mining, water bodies, wasteland, and fallow land. The classifiers were then used to regroup the land into four categories: vegetation (including forest and agricultural land), built-up (including built-up and mining areas), water bodies, and others (wasteland and fallow land), which were then exported. As the next step, the mining and forest masks were manually prepared and applied. The final consolidated classes for modeling were agricultural, built-up, others (wasteland and fallow land), and restricted (water bodies, mining, and forest). In modeling the predicted growth, the class "restricted" will serve as a constraint for urban sprawl.

*4.3. Accuracy Assessment*

The accuracy of LU/LC datasets was validated by sampling. First, the optimal number of samples for each class was calculated using Yamane's formula (Equation (1)), [46].

$$n = \frac{N}{1 + N(e)^2} \tag{1}$$

where $n$ represents the sample size, $N$ is the total population, and $e$ denotes the allowed margin of error. A stratified random sampling technique was then used to allocate the samples spatially. The sampling was performed using the "random points in the layer bounds" tool in QGIS for each feature separately, as the number of points for the built-up class was different from the remaining classes. A confusion matrix was then created to compare the actual and predicted classes.

Several statistical measures were developed to evaluate the accuracy of the classifications. Overall accuracy and error rate are standard measures for classification results across classes. Accuracy and error rate are complementary. Furthermore, the error rate portrays the test set rather than the entire population. Kappa coefficient is an alternative method for defining classification agreement [47,48]. Accuracy, error rate, and kappa coefficient ($k$) can be determined using the formulas given in Equations (2)–(4).

$$Accuracy = \frac{TP + TN}{All} \tag{2}$$

$$Error\ rate = (1 - Accuracy) \tag{3}$$

$$\kappa = \frac{2 \times (TP * TN - FN * FP)}{(TP + FP) * (FP + TN) + (TP + FN) * (FN + TN)} \tag{4}$$

*TP*, *FP*, *TN*, and *FN* represent true positives, false positives, true negatives, and false negatives, respectively. *TP* and *TN* mean that the presence and absence of class are correctly classified. *FP* denotes wrongly classified pixels in a class, and *FN* denotes those pixels of the class that are misclassified into another class.

To assess class performance, we employ additional measures such as recall (or sensitivity), precision, specificity, and the F-score or F-measure [49–51]. Precision and recall are measures of incorrect classification and their associated misclassification errors (Equations (5) and (6)). The F-measure is a score to assess both in a single variable (Equation (7)). Equation (8) [52] provides a formula for calculating specificity, a measure of the total negative recognition rate.

$$Recall\ or\ sensitivity = \frac{TP}{TP + FN} \tag{5}$$

$$Precision = \frac{TP}{TP + FP} \tag{6}$$

$$F - measure = 2 * \frac{Precision * recall}{Precision + recall} \tag{7}$$

$$Specificity = \frac{TN}{TN + FP} \tag{8}$$

In addition, we graph the receiver operating characteristics (ROC) curve, plotting true positive rates (TPR) and false positive rates (FPR) from Equation (9) and calculating the area under the curve. The area under the curve that equals 1 is the ideal classification result. The sensitivity is the true positive rate, and the true negative rate is the complement of specificity and is thus calculated as

$$FPR = 1 - Specificity \tag{9}$$

Thus, the classified images of 2001, 2011, and 2020 were evaluated for accuracy.

### 4.4. Change Detection and Urban Growth Analysis

Change detection is required as a manifestation to infer that sufficient sprawl has occurred to define urban sprawl modeling. This ensures the study's validity and helps to understand how LU/LC classes influence the spatial occurrence of urban sprawl. The classified images were used to compute the magnitude and rate of change. A change detection matrix was used to detect changes. The primary advantage of post-classification comparison is that the dates of images are classified separately, reducing the problem of systematic and non-systematic remote sensing errors.

The rate of change of urban sprawl is calculated by dividing the area of each LU/LC class into two-time slices [53]. The formula for calculating the rate of change is provided in Equation (10).

$$p = \frac{1}{(t2 - t1)} \frac{(a2 - a1)}{a1} \tag{10}$$

where p denotes the annual rate of change (percentage per year), and $a1$ and $a2$ represent the area of the LU/LC classes at times $t1$ and $t2$, respectively. Finally, the change in urban extent is observed using the computed values, and inferences were made.

### 4.5. Hybrid CA–Markov Modeling

The Markov chain process, a machine learning approach that models the LU/LC change on various scales with several assumptions, serves as the foundation of the CA–Markov model [24]. One of the assumptions is that changes in LU/LC are a stochastic process and that the various classes in LU/LC represent the state at any given time. When predicting how a variable changes over time, the Markov model considers previous states. The model produced the probabilities of the states of conversion between each LU/LC. Equations (11) and (12) show the mathematical representation of the Markov model [54]:

$$L_{(t+1)} = P_{ij} * L_{(t)} \tag{11}$$

and

$$P_{ij} = \begin{bmatrix} P_{11} & P_{12} & \dots & P_{1m} \\ P_{21} & P_{22} & \dots & P_{2m} \\ \dots & \dots & \dots & \dots \\ P_{m1} & P_m & \dots & P_{mm} \end{bmatrix} \tag{12}$$

where $L_{(t)}$ and $L_{(t+1)}$ are the LU/LC status at periods t + 1 and t.

$$0 \le P_{ij} < 1 \text{ and } \sum_{j=1}^{m} P_{ij} = 1$$

where ($j$ = 1, 2, . . . ,m) is the transition probability matrix in a state. Cross-tabulation of two LU/LC maps of different times yields a transition probability matrix. It calculates the

likelihood of a pixel in one LU/LC class transitioning to another during that period. On the other hand, a transition area matrix contains the number of pixels that are projected to shift from one class to another within a certain period. Thus, the model creates a transition probability and transition area matrix. Once the transition probability is computed, the CA–Markov model can predict LU/LC [55,56].

This model requires inputs, namely a base year image for which projection is to be performed, a transition area file derived from the Markov model, a transition suitability image, and a filter. The transition suitability image was generated using the CellAtom module in Terrset. Based on the particular transition rule, a self-reproductive cell on the grid space will assume a finite number of alternative cell states, which may interact with the states of its immediate neighbors on the same grid space. Specific transition rules that regulate changes in the cell state can be used to forecast the changes. The module requires two inputs: a reclass file (stating the possible transitions) and a filter. A contiguity filter was used for both analyses.

To produce spatially distributed continuous weighing factors, a kernel size of $5 \times 5$ that accounts for the neighborhood pixels was chosen, as shown in Figure 3. Pixels further away from the existing LU/LC class were deemed less suitable than pixels closer to the existing LU/LC class [57,58]. The present study used Markov chain and CA–Markov models available in Terrset. The Markov model was used to develop a transition matrix, and CA–Markov was used for prediction. Under existing socioeconomic conditions, it is assumed that the LU/LC changes observed between 2001 and 2011 will continue in the future. As a result, the transition potential matrix in this scenario was computed using annual transition probabilities from 2001 to 2011. The model was then configured to run from 2011 to 2020 (base year). Finally, the model's accuracy was validated by comparing the simulated LU/LC (2020) to the actual LU/LC (2020). The transition matrix can be used to model the measurement of change in each transition. Briefly, the model assesses how variables influence future change, measures the difference between 2001 and 2011, and estimates a relative transition to 2030.

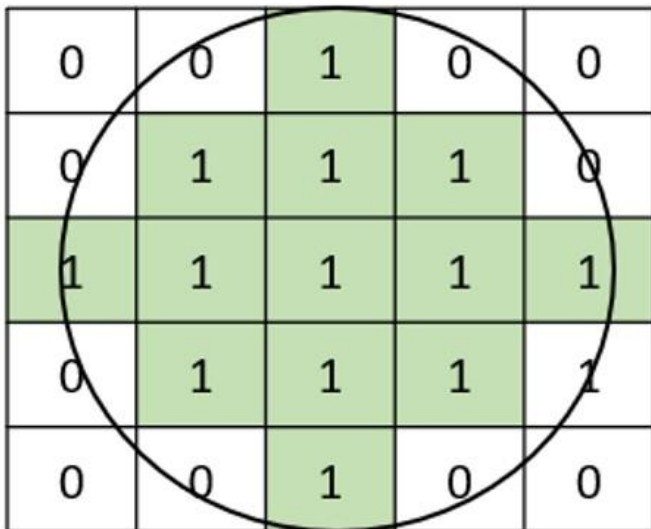

**Figure 3.** A $5 \times 5$ filter defines the spatial influence of neighboring cells.

## 5. Results and Discussions

### 5.1. Classification and Accuracy Assessment

Pan-sharpened Landsat 7 and Landsat 8 data were used to classify LU/LC by performing the support vector machine algorithm in Google Earth Engine. Finally, the output was exported, and masks were applied. Final LU/LC classes of 2001, 2011, and 2020 obtained through the analysis of multi-temporal satellite images were illustrated as thematic maps in Figures 4–6.

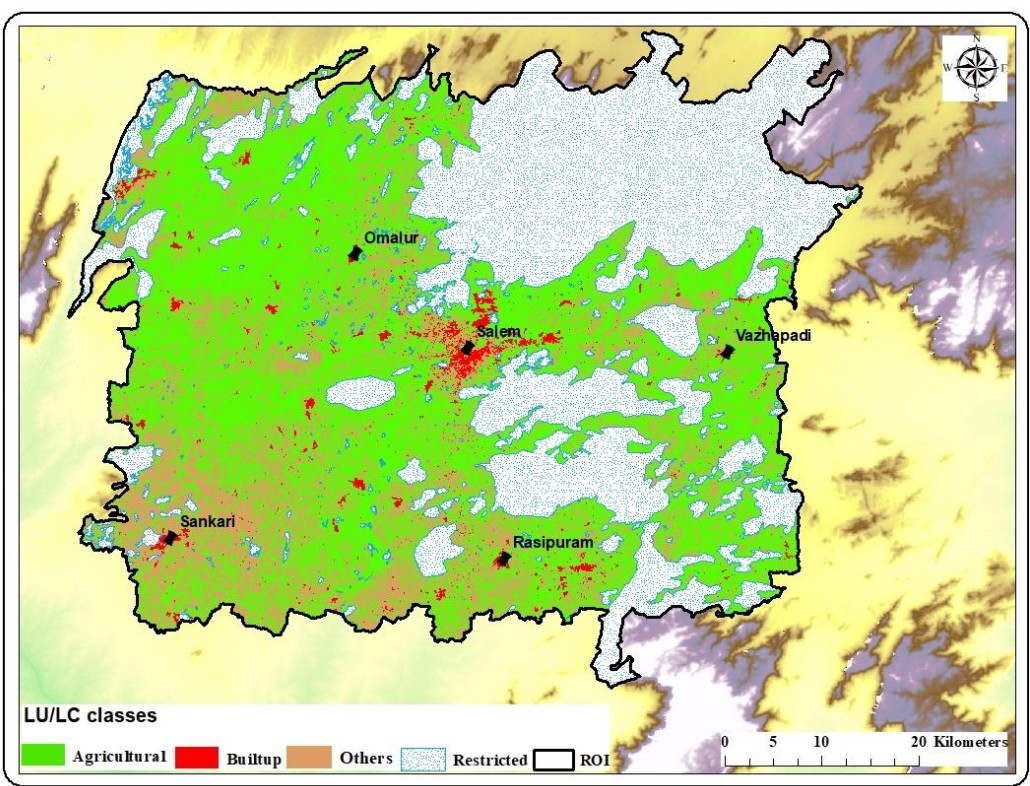

**Figure 4.** LU/LC classes in the year 2001.

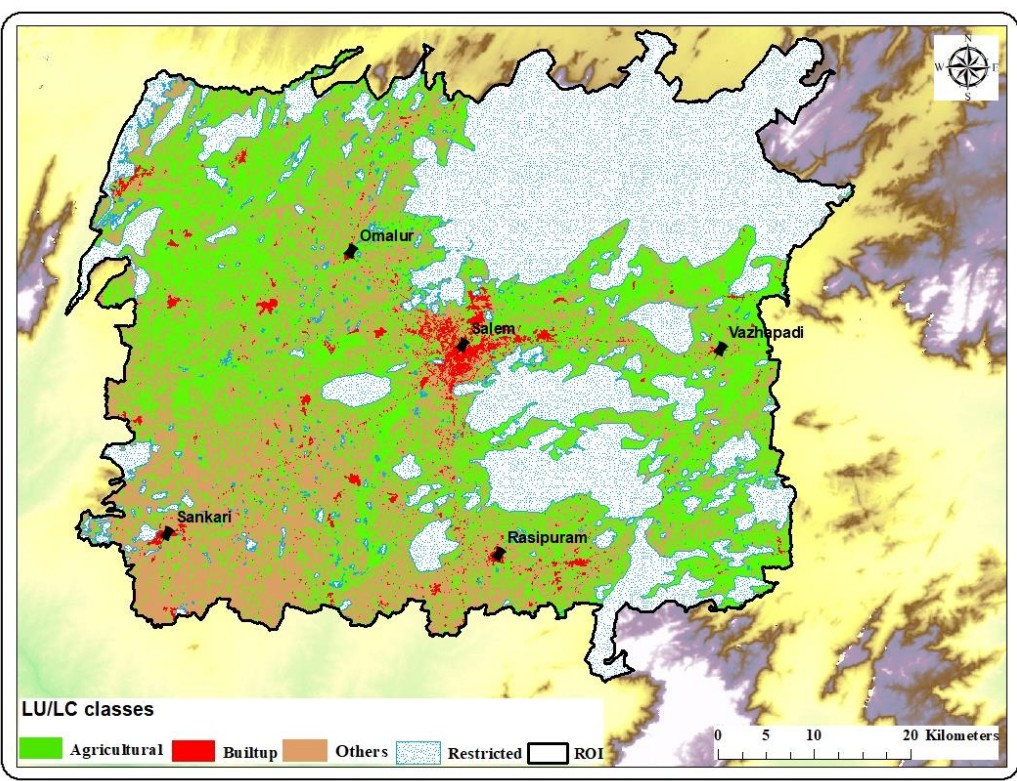

**Figure 5.** LU/LC classes in the year 2011.

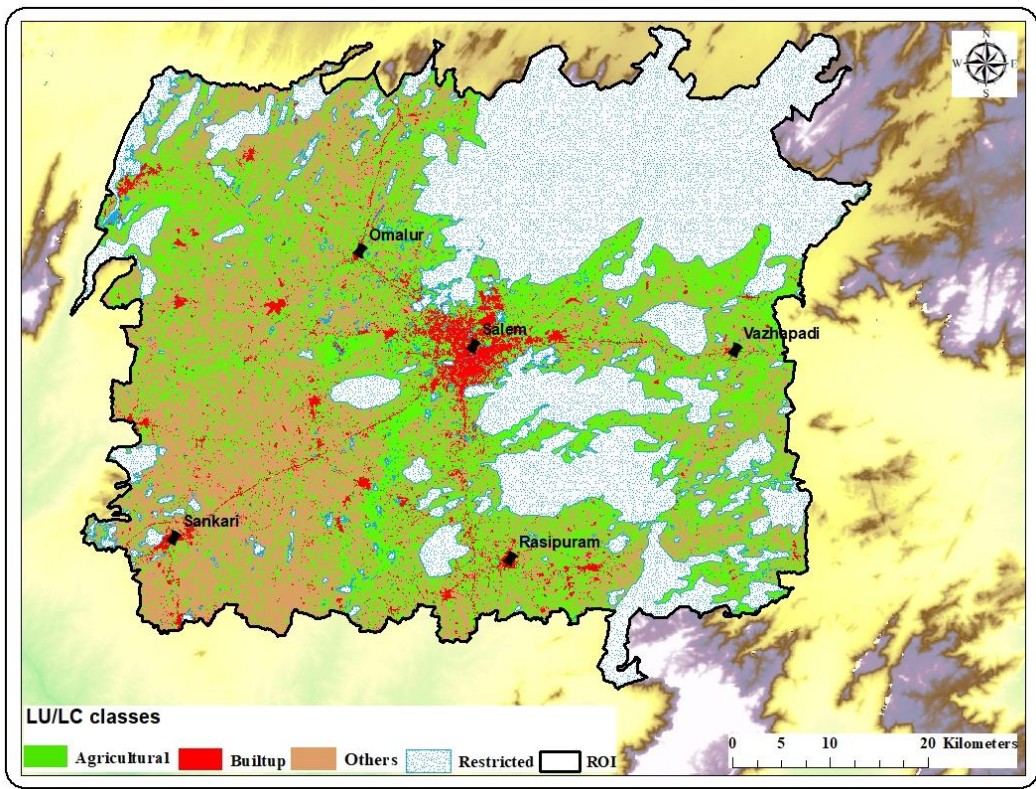

**Figure 6.** LU/LC classes in the year 2020.

As the first step for accuracy assessment, the optimum samples were calculated. In the Yamane formula, we assumed the margin of error as 8% for built-up and 10% for the remaining classes to give more weightage for built-up. The optimum number of samples for all three years was determined as follows: Vegetation = 100, Built-up = 150, Others = 100, and Restricted = 100. Accuracy assessment was performed for all three LU/LC maps using their corresponding high-resolution satellite images, Google Earth, and ground checks as reference. It was found that the overall accuracies were 91.4, 92.3, and 95.2%, and kappa statistics were 0.884, 0.896, and 0.935 for 2001, 2011, and 2020, respectively. The remaining statistics are discussed in Tables 2–4, and the ROC curves are shown in Figure 7. The area under the ROC curve was calculated and found as 0.942, 0.950, and 0.976 for 2001, 2011, and 2020. These measures prove that the LU/LC maps are accurate for further analysis.

**Table 2.** Results of accuracy assessment—2001.

| | Vegetation | Built-Up | Others | Restricted |
|---|---|---|---|---|
| **Recall/Sensitivity** | 0.890 | 0.896 | 0.910 | 0.970 |
| **Precision** | 0.918 | 0.939 | 0.805 | 1.000 |
| **Specificity** | 0.921 | 0.923 | 0.915 | 0.898 |
| **F1 Score/F Measure** | 0.904 | 0.917 | 0.854 | 0.985 |
| **FPR** | 0.079 | 0.077 | 0.085 | 0.102 |
| **TPR** | 0.890 | 0.896 | 0.910 | 0.970 |
| **Kappa Statistics** | | 0.884 | | |
| **Overall Accuracy** | | 91.4 | | |
| **Error Rate** | | 0.086 | | |

**Table 3.** Results of accuracy assessment—2011.

|  | Vegetation | Built-Up | Others | Restricted |
|---|---|---|---|---|
| **Recall/Sensitivity** | 0.930 | 0.896 | 0.910 | 0.970 |
| **Precision** | 0.921 | 0.939 | 0.835 | 1.000 |
| **Specificity** | 0.921 | 0.937 | 0.927 | 0.910 |
| **F1 Score/F Measure** | 0.925 | 0.917 | 0.871 | 0.985 |
| **FPR** | 0.079 | 0.063 | 0.073 | 0.090 |
| **TPR** | 0.930 | 0.896 | 0.910 | 0.970 |
| **Kappa Statistics** | | 0.896 | | |
| **Overall Accuracy** | | 92.3 | | |
| **Error Rate** | | 0.077 | | |

**Table 4.** Results of accuracy assessment—2020.

|  | Vegetation | Built-Up | Others | Restricted |
|---|---|---|---|---|
| **Recall/Sensitivity** | 0.930 | 0.922 | 0.980 | 0.990 |
| **Precision** | 0.959 | 0.993 | 0.852 | 1.000 |
| **Specificity** | 0.958 | 0.967 | 0.944 | 0.941 |
| **F1 Score/F Measure** | 0.944 | 0.956 | 0.912 | 0.995 |
| **FPR** | 0.042 | 0.033 | 0.056 | 0.059 |
| **TPR** | 0.930 | 0.922 | 0.980 | 0.990 |
| **Kappa Statistics** | | 0.935 | | |
| **Overall Accuracy** | | 95.2 | | |
| **Error Rate** | | 0.048 | | |

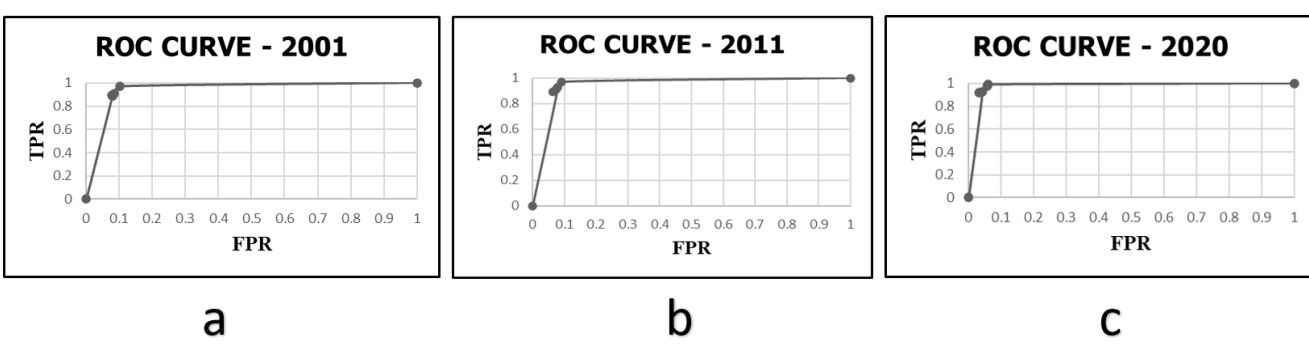

**Figure 7.** ROC curve for (**a**) 2001, (**b**) 2011, and (**c**) 2020.

*5.2. Time Series Analysis of LU/LC and Urban Growth*

The support vector machine algorithm in Google Earth Engine was used to classify LU/LC using pan-sharpened Landsat 7 and Landsat 8 data. The output was then exported, and masks were applied. Final LU/LC classes of 2001, 2011, and 2020 obtained from multi-temporal satellite images are shown in Figures 4–6. The data are shown in Tables 4 and 5. Urban sprawl can be quantified by comparing time series change over three years. Consequently, the changes between each time slice (2001–2011 and 2011–2020) were investigated by intersecting the respective LU/LC layers. The percentage of the difference between classes was then computed (Tables 5 and 6).

**Table 5.** LU/LC change matrix (2001–2011) (in %).

| | LU/LC Categories | Year 2001 | | | |
|---|---|---|---|---|---|
| | | Vegetation | Built-Up | Others | Restricted |
| Year 2011 | Vegetation | 80.01 | 0.07 | 19.77 | 0.15 |
| | Built-up | 9.09 | 69.82 | 20.93 | 0.15 |
| | Others | 53.35 | 0.42 | 46.15 | 0.08 |
| | Restricted | 0.28 | 0.01 | 0.47 | 99.24 |
| | Class total | 100.00 | 100.00 | 100.00 | 100.00 |

**Table 6.** LU/LC change matrix (2011–2020) (in %).

| | LU/LC Categories | Year 2011 | | | |
|---|---|---|---|---|---|
| | | Vegetation | Built-Up | Others | Restricted |
| Year 2020 | Vegetation | 68.45 | 0.86 | 30.44 | 0.25 |
| | Built-up | 12.34 | 37.07 | 49.9 | 0.69 |
| | Others | 38.72 | 1.39 | 59.65 | 0.25 |
| | Restricted | 0.28 | 0.04 | 0.12 | 99.56 |
| | Class total | 100.00 | 100.00 | 100.00 | 100.00 |

Tables 5 and 6 show that the majority of the "others class" (barren land, wasteland, and cropland) have been converted into built-up land. Similarly, the vegetation cover has been transformed into built-up land. Thus overall, the two-time slices have shown a steadily rising trend, indicating the presence of unchecked urban growth in the area. The annual rate of change represents the average annual conversion rate of LU/LC classes per unit area [59]. From Tables 7 and 8, a decline in LU/LC can be inferred from the negative rate of change in vegetation between 2001–2011 and 2011–2020. On the other hand, an increase is observed in the "others class" during the above time slices. However, the built-up cover, in contrast, demonstrated the smallest increase. The vastness of the study area could be a plausible explanation for the low representation. Furthermore, under Indian law, fertile land cannot be converted to be used for residential purposes. Only parcels of dry or barren land could be converted. Hence, it can be presumed that the vegetation cover has been changed to "other class" and that it will be converted to "built-up" in the near future (Table 8). The transition rate is expected to accelerate in the near future, especially after the implementation of the Salem smart city mission.

**Table 7.** Area and rate of change (per year) of different LU/LC classes (2001–2011).

| LU/LC Categories | 2001 | | 2011 | | Change in Area (2001–2011) | Rate of Change |
|---|---|---|---|---|---|---|
| | Km$^2$ | % | Km$^2$ | % | Km$^2$ | % per Year |
| Vegetation | 1648.080 | 42.248 | 1277.458 | 32.747 | −370.622 | −0.022 |
| Built-up | 59.606 | 1.528 | 76.922 | 1.972 | 17.316 | 0.029 |
| Others | 807.323 | 20.695 | 1153.024 | 29.557 | 345.701 | 0.043 |
| Restricted | 1386.001 | 35.529 | 1393.605 | 35.724 | 7.604 | 0.001 |
| Total | 3901.010 | 100.000 | 3901.010 | 100 | – | – |

**Table 8.** Area and rate of change (per year) of different LU/LC classes (2011–2020).

| LU/LC Categories | 2011 | | 2020 | | Change in Area (2011–2020) | Rate of Change |
|---|---|---|---|---|---|---|
| | Km² | % | Km² | % | Km² | % per Year |
| Vegetation | 1277.4583 | 32.747 | 1135.595 | 29.110 | −141.863342 | −0.012 |
| Built-up | 76.922223 | 1.972 | 133.300 | 3.417 | 56.378252 | 0.081 |
| Others | 1153.0242 | 29.557 | 1239.226 | 31.767 | 86.202187 | 0.008 |
| Restricted | 1393.6052 | 35.724 | 1392.888 | 35.706 | −0.717099 | 0.000 |
| Total | 3901.010 | 100 | 3901.010 | 100 | – | – |

### 5.3. CA–Markov Modeling

The model was trained using data from 2001 and 2011 and projected for 2020. Initial execution of the Markov model provides the 2001–2011 transition area and probability matrices. Tables 9 and 10 depict that the maximum transition that occurs from vegetation to others with a probability of 0.3640. Although restricted classes provide constraints in LU/LC change modeling, it was realized that some transitions from the restricted to remaining classes are possible but with insignificant probabilities. Because water bodies are one of the classes included in the restricted class and can become covered by algae deposition, dry up and act as barren land, or be covered by built-up structures, the aforementioned transition is possible. There was also a transition from others to vegetation observed with a probability of 0.3056, which is the result of fallow land in the first year being mapped as "others", being covered by cropland in the following year, and later being mapped as "vegetation". Since we are concentrating on urban sprawl, this will not have any direct impact on the results of the analysis.

**Table 9.** Transition area matrix for 2001–2011.

| Cells in: 15 m | Expected to Transition to | | | |
|---|---|---|---|---|
| | Vegetation | Built-Up | Others | Restricted |
| Vegetation | 3,579,819 | 18,913 | 2,066,187 | 10,857 |
| Built-up | 1225 | 363,688 | 1225 | 1225 |
| Others | 1,559,392 | 95,489 | 3,409,269 | 38,246 |
| Restricted | 48,375 | 2828 | 22,338 | 6,118,712 |

**Table 10.** Transition probability matrix for 2001–2011.

| Given | Probability of Changing To | | | |
|---|---|---|---|---|
| | Vegetation | Built-Up | Others | Restricted |
| Vegetation | 0.6307 | 0.0033 | 0.3640 | 0.0019 |
| Built-Up | 0.0033 | 0.9900 | 0.0033 | 0.0033 |
| Others | 0.3056 | 0.0187 | 0.6682 | 0.0075 |
| Restricted | 0.0078 | 0.0005 | 0.0036 | 0.9881 |

Suitability images were created as the next step in the modeling process, using reclass and filter, which define which class will be converted to which class and how much neighboring cells will influence this change spatially. Finally, using the matrices and suitability files, the CA–Markov model was used to forecast 2020. (Figure 8). The observed transition probability of 0.3056 from the other classes to the vegetation class in the predicted image demonstrates that the model significantly underestimates the quantity of vegetation present (Table 10). Similar findings were discussed by [60,61], who found that the model underestimated agricultural land. As previously stated, because the trade-off is only between vegetation and other classes, it will have no effect on the prognosis of urban sprawl.

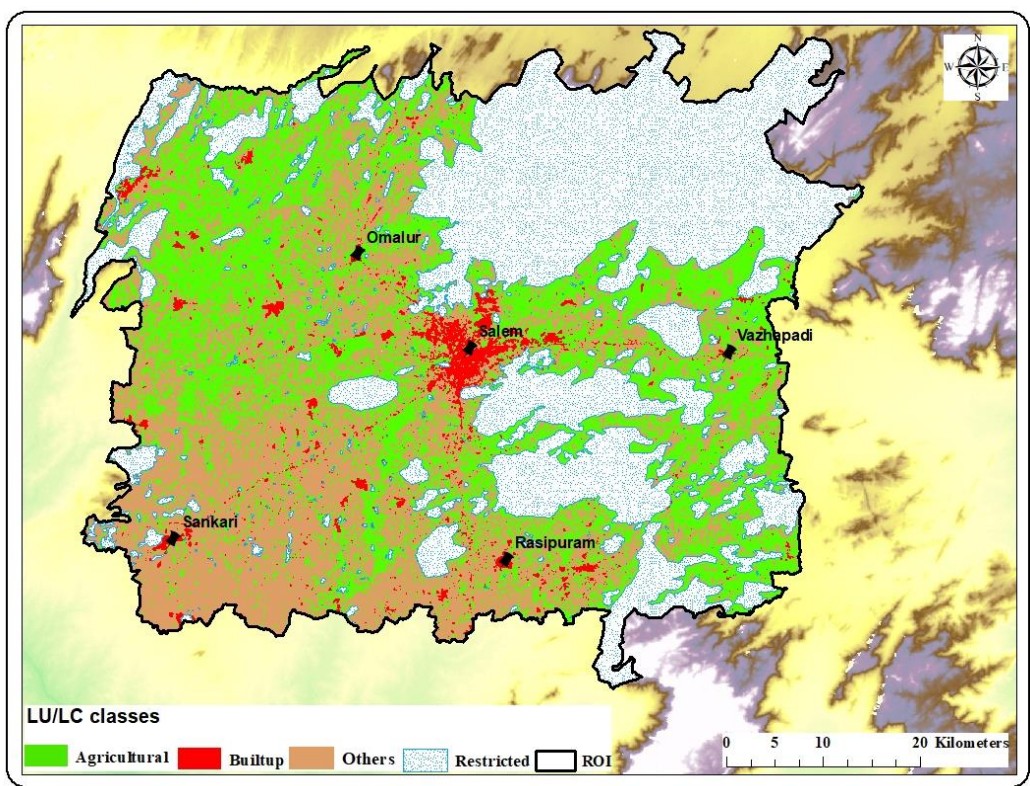

**Figure 8.** LU/LC classes predicted for the year 2020.

*5.4. Model Validation*

Validation is essential for any machine learning prediction algorithm. The IDRISI Terrset includes a VALIDATE module used for evaluating the results. To determine the accuracy of the models, we first compare the reference (actual) and predicted maps. The filter file is modified to generate suitable images if the results are deemed insufficient. Several types of kappa statistics exist, including the Kappa standard (Kstandard), Kappa for no information (Kno), Kappa for location (Klocation), and Kappa for location strata (Klocationstrata), which can be computed using the VALIDATE tool in Terrset. When kappa is greater than 0.6, actual and predicted classes agree perfectly [62]. With a 5 × 5 filter, the outcomes of the model were validated, and the kappa statistics are shown in Table 11. It can be inferred that values were higher than 0.75, suggesting that the model is statistically significant. These results are similar to the ANN-CA–Markov model with driver variables [49]. The comparison of results demonstrates that the method used in this paper is highly satisfactory for non-metropolitan Indian cities such as Salem, even without driver variables.

**Table 11.** Kappa values for predicted LU/LC for the year 2020.

| Kappa Statistics | Values |
|:---:|:---:|
| Kstandard | 0.7734 |
| Kno | 0.7861 |
| Klocation | 0.7811 |
| KlocationStrata | 0.7811 |

*5.5. Prediction of Urban Sprawl*

Following the validation and acceptance of the model, the CA–Markov model is used to predict the LU/LC for 2030 using the same reclass and filter file assumptions. The Markov model was applied to determine the transition area and potential matrices for 2020–2030 (Tables 12 and 13). According to the transition probability matrix, classes

such as vegetation, built-up, and others are transitory categories that are subject to more changes over time. Vegetative and other areas (primarily fallow, barren, and wasteland) shifted primarily to built-up areas. The suitability file for 2020 was then created to train the model and to understand the changes from 2020. Finally, the CA–Markov model was run with 2020 LU/LC as the base image and a time step of 10 years to forecast the year 2030 (Figure 9).

**Table 12.** Transition area matrix for 2011–2020.

| Cells in: 15 m | Expected to Transition to | | | |
|---|---|---|---|---|
| | Vegetation | Built-Up | Others | Restricted |
| **Vegetation** | 2,705,764 | 63,320 | 2,219,952 | 16,728 |
| **Built-up** | 11,128 | 672,256 | 24,633 | 1811 |
| **Others** | 1,930,618 | 344,750 | 3,150,912 | 8436 |
| **Restricted** | 39,383 | 12,359 | 42,345 | 6,093,394 |

**Table 13.** Transition probability matrix for 2011–2020.

| Given | Probability of Changing to | | | |
|---|---|---|---|---|
| | Vegetation | Built-Up | Others | Restricted |
| **Vegetation** | 0.5405 | 0.0126 | 0.4435 | 0.0033 |
| **Built-Up** | 0.0157 | 0.9471 | 0.0347 | 0.0026 |
| **Others** | 0.3552 | 0.0634 | 0.5798 | 0.0016 |
| **Restricted** | 0.0064 | 0.0020 | 0.0068 | 0.9848 |

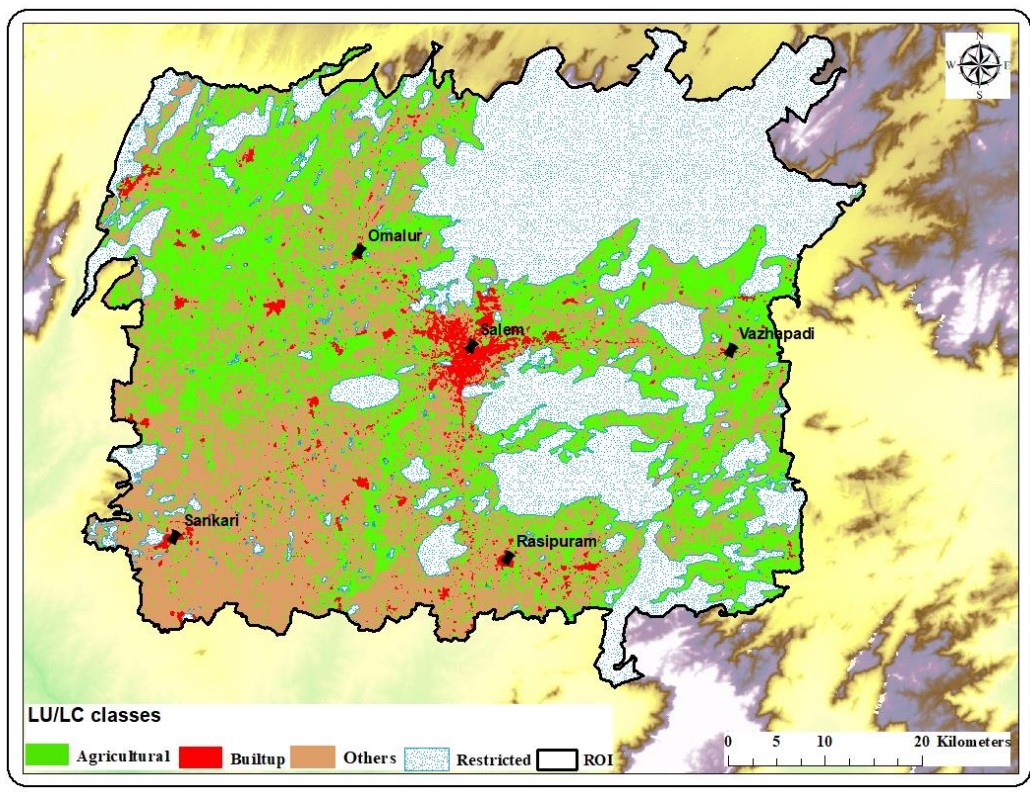

**Figure 9.** LU/LC classes predicted for the year 2030.

Table 14 summarizes the area of each class from 2001 to 2030. It is reasonable to conclude that a consistent decline in vegetation cover results in the loss of cropland and plantations. These covers may or may not be developed into a built-up structure, but they are more likely turned into barren or fallow land. However, these covers are susceptible to

urbanization in the years to come. Only for modeling purposes are the mining, forested, and water body classes consolidated into the restricted class. From 2001 to 2020, it was seen that the mining areas in the northwest expanded steadily. This serves as one of the pull factors for urban sprawl around Salem city. The sprawl can also be seen in the neighboring towns of Omalur, Rasipuram, Sankari, and Vazhapadi. The proximity between Salem and these towns, as well as the existence of a connection between them, will promote future urban expansion.

**Table 14.** Actual (2001, 2011, and 2020) and predicted area of LU/LC for 2030 (sq. km).

| LU/LC Class | 2001 (Actual) | 2011 (Actual) | 2020 (Actual) | 2020 (Predicted) | 2030 (Predicted) |
|---|---|---|---|---|---|
| Vegetation | 1648.080 | 1277.458 | 1135.595 | 1055.351 | 1080.189 |
| Built-up | 59.606 | 76.922 | 133.300 | 244.483 | 179.638 |
| Others | 807.323 | 1153.024 | 1239.226 | 1223.791 | 1262.751 |
| Restricted | 1386.001 | 1393.605 | 1392.888 | 1377.385 | 1378.432 |
| Total | 3901.010 | 3901.010 | 3901.010 | 3901.010 | 3901.010 |

## 6. Conclusions

The study investigated the spatial–temporal pattern of LULC change, particularly the urban expansion dynamics of Salem city, India, with the aid of Landsat multitemporal imageries, from 2001 to 2020, and predicted future changes by 2030 using the CA–Markov chain model. The analysis explored the remarkable increase in barren land and a significant decline in the agricultural area. Especially, the built-up area has increased from 59.6 sq km in 2001 to 76.9 sq km in 2011 and 133.3 sq km in 2020. The modeling results also predicted a continued growth to 179.6 sq km by 2030. Thus, it is reasonable to conclude that urban sprawl in Salem and the neighboring towns is steadily increasing (Omalur, Rasipuram, Sankari, and Vazhapadi). Even though the increase in urban land is marginal, the conversion of agricultural land to barren land is prominent. As previously discussed, these barren lands would eventually be developed into built-up areas in the years to come. As the city has enormous development potential due to its employment opportunities and further as it becomes a smart city, the transitions from agricultural and barren/wasteland to built-up areas will proliferate exponentially. The expansion of mining areas sends a warning message not only about urban sprawl but also about environmental degradation.

In this study, the spatial changes have been examined in terms of how the land area is being impacted by cells that are located in close proximity to one another. The validation of these underlying principles has shown satisfactory accuracy. Earlier studies on urban sprawl that included driver variables have yielded similar prediction accuracy. Therefore, the proposed methodology can be used to study urban sprawl with acceptable precision even when driver variables are unavailable. The spatial understanding of sprawl through the decades and from prediction shows that the sprawl of the neighboring towns is more prominent in the direction aligned toward Salem except for Omalur. The sprawl in Omalur is projected in the northwest direction, but Salem's growth rate is high in the northwest, which is toward Omalur. These findings emphasize the fact that in the future, any of these towns could merge with Salem city. Hence, urban planners and policymakers must also consider this as a perspective when planning or improvising infrastructural facilities for Salem city. The urban sprawl modeled and predicted in this study will be helpful in understanding morphological changes and estimating the city's population growth. Policymakers could use the study's findings to develop appropriate urban management plans. Salem has the potential to be a well-planned smart city if careful measures are taken and sustainable management protocols are implemented.

The approach of implementing the CA–Markov model without the driver variables has its own advantage, as it can provide a baseline scenario for urban growth and change, which can then be compared with scenarios that include driver variables. This can reveal the relative importance of different factors and their interactions as urban growth drivers.

However, using CA–Markov models without driver variables has limitations, especially concerning accuracy and reliability. Furthermore, they may fail to capture the complexity and heterogeneity of urban processes and land-use changes, which are especially important in rapidly urbanizing cities. In terms of future scope, the proposed works are as follows: (a) incorporating driver variables such as population growth, income, and employment opportunities may improve the model's predictions; (b) including the impact of natural disasters and climate change on urban sprawl may provide a more comprehensive understanding of the phenomenon; (c) applying the model to different geographic regions and comparing the results may aid in identifying the factors that influence urban sprawl in various contexts; and (d) finally, because this temporal calculation of urban sprawl does not account for morphological changes, a directional study of urban sprawl and its pattern can be undertaken to comprehend the relationship between Salem and the neighboring towns.

**Author Contributions:** Conceptualization, S.R.; methodology, L.T.; software, L.T.; validation, S.R. and A.R.; formal analysis, L.T.; investigation, A.R.; resources, L.T.; data curation, L.T.; writing—original draft preparation, L.T.; writing—review and editing, S.R. and A.R.; visualization, L.T.; supervision, S.R. All authors have read and agreed to the published version of the manuscript.

**Funding:** This research received no external funding.

**Institutional Review Board Statement:** Not applicable.

**Informed Consent Statement:** Not applicable.

**Data Availability Statement:** The data presented in this study are available on request from the corresponding author. The data are not publicly available due to privacy reasons.

**Conflicts of Interest:** The authors declare no conflict of interest.

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
