# Peer review of "Simulating Urban Growth Using the Cellular Automata Markov Chain Model in the Context of Spatiotemporal Influences for Salem and Its Peripherals, India"

_2673-4834, doi:10.3390/earth4020016_

Round 1

Reviewer 1 Report

The presentation reflects the present state of knowledge. The paper is very well structured. The Introduction section is good, in this section the authors present clearly the objectives and the main contributions of the study. The authors provided sufficient background and include relevant references. The method is adequately described. The results are clearly presented. The conclusions are supported by the results. Authors should follow the format of the journal.

Reviewer 2 Report

I congratulate the author(s) for the work done. I think it is an interesting work. I believe it should be published and is relevant to the scientific community. However, for its better visibility and readability, I make constructive recommendations, in the best academic spirit.
The title is suggestive and appealing, and academically, it contains the summary of what the reader will find in the text.
The summary is good and attractive, and contains almost all the sections desirable in a summary (introduction-objectives-methodology-results-conclusions). However, for the sake of better indexing the article in databases, I think it would be good to add some more context.
The keywords are well chosen. But as they are almost all compound, perhaps I would recommend adding 2 or 3 simple words, before the ones that are already there, and that match more common terms such as UNESCO's. This will help better visibility of the article. This will help to improve the visibility of the article in databases, understanding that researchers do not always use these compound words and the words in the title usually already function as keywords.
In the Introduction and Conclusions, a clearer and more direct case should be made for why a useful methodology is being offered to the scientific community, why it is innovative or how an existing methodology has been updated, and how this methodology can be replicated by other researchers. This is not clearly stated and with what has already been written on similar issues, it is necessary to differentiate and delve into why this text offers something new.
There are hardly any citations in the context of the problem, nor in the state of the art. Although the texts cited are very solid and reputable, some are a bit old for the field and there are few recent and international citations for a publisher of this prestige and for this topic. I recommend adding 3-4 super up-to-date references, from top international journals or publishers, especially when an article on this topic is being offered.
The section on Results is excellent, as the exposition and argumentation are very well presented and well spun. It is the best part of the text and I congratulate you on the exposition. The tables and figures are very readable, appropriate and perfectly set out the results.
All acronyms should have their first letter in capital letters, at least the first time they appear.
A harmonious distribution of paragraphs should be sought, so that they all have a similar length, 6-7 lines, without very long paragraphs. This will make the text more readable and understandable, even if it is already well written.
It would be good to include, at the end, an expansion of the limitations encountered and the prospects proposed for the research community. This is the most valuable part of research on this topic.

Reviewer 3 Report

The study provides valuable insights into the spatiotemporal dynamics of land use/land cover changes and urban growth patterns in Salem and its surrounding communities. The authors employed cellular automata (CA)-Markov and geospatial techniques to simulate urban expansion in 2030. The study's outcomes can serve as spatial guidelines for growth regulation and monitoring, which is particularly important given the rapid urbanization and sprawl in Salem and the surrounding towns.

(1) The study can provide a more comprehensive review of the application of CA-Markov in urban studies and highlight the research gaps. Additionally, the authors could provide more information about the methods used to analyze the spatiotemporal dynamics of land use/land cover changes, such as the data sources, spatial resolution, and accuracy assessment. This information could help readers better evaluate the study's reliability and validity.

(2) The authors are highly encouraged to add a paragraph to provide important background information about the study area, which could enhance the readers' understanding of the study's context and significance. 

Overall, the study's findings are significant for policymakers and planners in Salem and the surrounding communities, as they provide valuable information for growth regulation and monitoring. However, more detailed information about the study area and methods would enhance the study's impact and value to the academic community
